# Research on Adaptive 1DCNN Network Intrusion Detection Technology Based on BSGM Mixed Sampling

**DOI:** 10.3390/s23136206

**Published:** 2023-07-06

**Authors:** Wei Ma, Chao Gou, Yunyun Hou

**Affiliations:** School of Information Engineering, North China University of Water Resources and Electric Power, Zhengzhou 450046, China; mawei@ncwu.edu.cn (W.M.); x20211090823@stu.ncwu.edu.cn (Y.H.)

**Keywords:** network intrusion detection, Gaussian mixture model, quantum particle swarm algorithm, mixed sampling

## Abstract

The development of internet technology has brought us benefits, but at the same time, there has been a surge in network attack incidents, posing a serious threat to network security. In the real world, the amount of attack data is much smaller than normal data, leading to a severe class imbalance problem that affects the performance of classifiers. Additionally, when using CNN for detection and classification, manual adjustment of parameters is required, making it difficult to obtain the optimal number of convolutional kernels. Therefore, we propose a hybrid sampling technique called Borderline-SMOTE and Gaussian Mixture Model (GMM), referred to as BSGM, which combines the two approaches. We utilize the Quantum Particle Swarm Optimization (QPSO) algorithm to automatically determine the optimal number of convolutional kernels for each one-dimensional convolutional layer, thereby enhancing the detection rate of minority classes. In our experiments, we conducted binary and multi-class experiments using the KDD99 dataset. We compared our proposed BSGM-QPSO-1DCNN method with ROS-CNN, SMOTE-CNN, RUS-SMOTE-CNN, RUS-SMOTE-RF, and RUS-SMOTE-MLP as benchmark models for intrusion detection. The experimental results show the following: (i) BSGM-QPSO-1DCNN achieves high accuracy rates of 99.93% and 99.94% in binary and multi-class experiments, respectively; (ii) the precision rates for the minority classes R2L and U2R are improved by 68% and 66%, respectively. Our research demonstrates that BSGM-QPSO-1DCNN is an efficient solution for addressing the imbalanced data issue in this field, and it outperforms the five intrusion detection methods used in this study.

## 1. Introduction

The widespread adoption of internet technology has brought tremendous convenience to our production and daily lives. However, in recent years, multiple targeted network attack incidents have indicated that while internet connectivity offers benefits and convenience, it also comes with significant cybersecurity risks [1]. Particularly in fields such as healthcare, aerospace, and automotive, if they are infiltrated and attacked by cybercriminals, not only can information be compromised, but people’s lives may also be endangered. As internet technology continues to evolve and iterate, various new network attack techniques continue to emerge, posing serious threats to cybersecurity. Therefore, network security has become even more crucial, and effective solutions must be implemented to defend against network attacks [2].

An Intrusion Detection System (IDS) is an essential proactive defense mechanism in network security and is considered the second line of defense after firewalls, playing a significant role in preventing and mitigating network attacks. In recent years, with the rapid development of deep-learning technology, intrusion detection methods combined with deep learning have made significant progress. From the perspective of data sources and detection techniques, intrusion detection systems can be classified into host-based intrusion detection, network-based intrusion detection, misuse-based intrusion detection systems, and anomaly-based intrusion detection. Compared to other intrusion detection methods, anomaly-based intrusion detection has become the focus of research due to its wide applicability, real-time capabilities, and the ability to detect unknown attacks.

In today’s world, the volume of data in the network environment is vast and complex. Machine learning and deep learning have the capability to learn from this data and discover hidden patterns and correlations, which is why they are applied in anomaly-based intrusion detection systems. Machine learning was initially used in intrusion detection research and has achieved impressive results. In the literature [3], a multi-layer filtering framework (MLFF) was proposed to perform feature dimensionality reduction on raw data, thereby improving detection time. In order to reduce the false positive rate in intrusion detection, the literature [4] introduced an OE-IDS model based on AutoML and soft voting. This model utilizes the AutoML framework to select the best-performing supervised classifier and employs a soft voting method to develop an optimized ensemble strategy, effectively reducing false positives. In the literature [5], an ant colony optimization algorithm was used for data dimensionality reduction, and support vector machines were used for classification. Experimental results on the KDD99 and NSLKDD datasets demonstrated the effectiveness of this method in improving accuracy. In recent years, with the rapid development of deep-learning technology, intrusion detection methods combining deep learning have also made significant progress. Deep-learning methods possess powerful feature extraction capabilities and can handle massive high-dimensional data [6,7,8]. Yin et al. [9] proposed a multi-scale convolutional neural network called M-CNN for IoT security. This model incorporates long short-term memory networks into M-CNN to enhance local feature extraction capabilities and utilizes the Inception network architecture as the backbone network to construct a multi-scale convolutional neural network. Experimental results on the KDD99 dataset achieved an accuracy of 93.90%. In the literature [10], stacked autoencoders were combined with multi-layer perceptron. The stacked autoencoders were used to reduce features, and the output of the autoencoders was used as the input for training the multi-layer perceptron. This method significantly improved the accuracy of detecting DoS attacks.

However, it is important to note that in the real world, the quantity of network attack traffic is far less than normal network traffic. This data imbalance issue affects the accuracy of intrusion detection models based on deep learning. Moreover, this problem is also evident in commonly used datasets in the field of intrusion detection. For example, in the UNSW-NB15 dataset, there are a total of 2,540,044 records, out of which 2,218,761 are labeled as normal, while the least represented class, worms, has only 174 records. In the CICIDS2017 dataset with 2,830,743 records, the BENIGN class has 2,273,097 records, while the least represented classes, Infiltration, Web Attack Sql Injection, and Heartbleed, have 36, 21, and 11 records, respectively. Combined, these three classes account for only 0.0024% of the total data. Addressing the issue of data imbalance is of significant importance for improving the accuracy and usability of intrusion detection methods. The problem of data imbalance has attracted the attention of many researchers, leading to the proposal of various solutions such as ROS and SMOTE. ROS is a simple undersampling method [11] that randomly discards a portion of the majority of class samples to balance the dataset. However, this method alters the original distribution of the data and can result in overfitting during experiments. SMOTE, on the other hand, is a representative oversampling method that addresses the issue of sample diversity but may generate overlapping samples. Existing approaches for dealing with class imbalance mainly suffer from issues such as limited sampling methods, overfitting, and sample overlap, failing to effectively address the class imbalance problem in intrusion datasets.

In the field of deep learning, the convolutional neural network (CNN) has achieved significant advancements in various domains. However, the traditional CNN requires manual parameter tuning during the training process, which has some drawbacks. Firstly, manual parameter tuning requires experienced experts who need to invest a considerable amount of time and effort in trying different parameter combinations. This can be a challenging and time-consuming task for beginners or researchers without extensive knowledge in the deep-learning domain. Secondly, the process of manual parameter tuning involves a certain level of subjectivity, as different experts may have different choices and preferences, leading to variations and inconsistencies in the results. Furthermore, due to the vast parameter space, manual parameter tuning is often limited to a finite set of parameter combinations, which may fail to fully explore the entire parameter space and potentially miss out on the optimal model configuration. Therefore, manual parameter tuning suffers from dependency on expert knowledge, subjectivity, and limitations, which restrict the automation and generalization capabilities of CNN. To address these issues, researchers are actively exploring methods for automated parameter tuning to improve the performance and efficiency of CNN.

Although deep-learning methods have made significant contributions to network intrusion detection, there are still some issues that affect the performance of deep-learning models. The main problems can be summarized as follows:The currently commonly used methods for handling class imbalance may lead to issues such as overfitting and data imbalance. These methods fail to address the problem effectively.When using CNN as a classifier, manual parameter tuning is required, making it difficult to find the global optimal solution.

To address the aforementioned issues, we propose a novel network intrusion detection model called BSGM-QOSO-1DCNN. Borderline-SMOTE [12] is a boundary-based oversampling algorithm that can improve classification performance, avoid generating noisy samples, and exhibit strong adaptability. The Gaussian Mixture Model is a non-linear clustering method that can model data distributions of arbitrary shapes. Therefore, it can be used for undersampling problems and better capture the complexity of the data. The Quantum Particle Swarm Optimization (QPSO) algorithm possesses strong global search capabilities and is employed to automatically find optimal parameters. Our model combines class imbalance handling, automatic parameter optimization, and deep learning to effectively address the issues of class imbalance and parameter optimization. We conduct binary and multi-class experiments on the KDD99 dataset and compare our approach with ROS-CNN, SMOTE-CNN, RUS-SMOTE-CNN, RUS-SMOTE-RF, and RUS-SMOTE-MLP. Our main contributions are as follows:We propose a new hybrid sampling technique called BSGM to address the issue of class imbalance. Firstly, we apply Borderline-SMOTE to oversample the minority class data in the KDD99 dataset. Secondly, we utilize the Gaussian Mixture Model (GMM) clustering algorithm to undersample the majority class data, resulting in a balanced dataset. This approach helps to avoid overfitting issues and effectively improves the detection rate of the minority class. In Section 3.2, the algorithm was described.We employ the QPSO algorithm to automatically obtain the parameters for the CNN. This eliminates the need for manual parameter tuning, overcoming the limitations of manual feature selection and avoiding wastage of time and computational resources. It enhances the performance and applicability of CNN. We described the process of optimizing CNN parameters using QPSO in Section 3.4.We evaluate the performance of ROS-CNN, SMOTE-CNN, RUS-SMOTE-CNN, RUS-SMOTE-RF, and RUS-SMOTE-MLP in binary and multi-class network intrusion detection. The experimental results demonstrate that our proposed BSGM-QOSO-1DCNN model outperforms the comparative models, providing an efficient solution for addressing class imbalance handling and parameter optimization. We extensively analyzed the experimental results in Section 4.The remaining sections of this paper are organized as follows. Section 2 discusses the related work on network intrusion detection. Section 3 provides a detailed description of the relevant techniques and the BSGM-QPSO-1DCNN model. Section 4 describes the parameter settings, evaluation metrics, and experimental results, followed by a discussion. Finally, Section 5 summarizes this research.

## 2. Related Works

With the development of computer technology, machine learning has been widely applied to intrusion detection [13,14,15,16,17,18] because it can uncover differential information between normal and malicious behaviors [19]. Saleh et al. [20] proposed a multi-class real-time intrusion detection system that utilizes the Naive Base Feature Selection (NBFS) technique to reduce the dimensionality of sample data. Subsequently, an Optimized Support Vector Machine (OSVM) is used to identify and eliminate outliers. Finally, the PKNN algorithm is employed for attack detection, and experimental results on multiple datasets validate the effectiveness of this approach. Chen et al. [21] introduced a novel network intrusion detection method that employs the Tree Seed Algorithm (TSA) for data processing, followed by classification using the K-Nearest Neighbor (KNN) classifier. Experimental results demonstrate that this combined model effectively removes redundant features and improves detection accuracy and efficiency. Shone et al. [22] proposed an intrusion detection model that combines an Asymmetric Multi-Layer Autoencoder with a Random Forest classifier, reducing computational costs and the required training data. Experimental results show that this model achieves a prediction accuracy of up to 97.85% on the KDD99 dataset.

In reference [23], a hybrid intelligent model combining Naive Bayes and Support Vector Machines (SVM) was proposed, which outperformed other methods in overall performance. For Multi-Protocol Label Switching (MPLS), reference [24] presented a machine-learning-based hybrid intrusion detection system that achieved 100% accuracy on the used dataset and improved time performance. Reference [25] combined the K-means algorithm with the XGBoost algorithm, utilizing K-means for processing raw data and XGBoost for efficient classification between normal and abnormal events. In reference [26], chi-square feature selection was employed to obtain the optimal features, optimizing decision trees for detection. The experiments conducted on the NSL-KDD, CICIDS2018, WSUTL, and ICS-SCADA datasets yielded the best performance.

Traditional machine learning has made certain contributions to network intrusion detection, but it is a shallow learning method. With the exponential growth of data, it becomes challenging to achieve the desired results. Deep learning, with its powerful feature extraction capability, has become a hot research topic in the field of network intrusion detection. Suda et al. [27] presented an algorithm for intrusion detection in vehicular networks that effectively extracts temporal features of data packets using Recurrent Neural Networks (RNNs). This algorithm captures the temporal patterns of data packets, enabling the detection of intrusion behavior. Reference [28] proposed an intrusion detection model based on Bidirectional Long Short-Term Memory (BiLSTM), training one LSTM on the original data and another on the reverse copy of the data. Compared to traditional models, this model improves the accuracy of U2R and R2L. Singla et al. [29] introduced an intrusion detection model based on Generative Adversarial Networks (GANs), which combines domain adaptation and GANs to achieve higher accuracy and precision with a small amount of training data while reducing training time. Liu et al. [30] proposed a method that utilizes GANs to address the imbalance and high dimensionality of the dataset. It generates minority class sample data using GANs and performs feature selection using analysis of variance, resulting in a balanced and low-dimensional dataset. This method effectively improves the accuracy of the model and addresses the problem of imbalanced datasets.

To address the issues of low accuracy and scalability in some existing intrusion detection methods, reference [31] proposes an improved Long Short-Term Memory (LSTM) approach, which outperforms the comparison models in multi-class detection. For the security threats faced by Industrial Control Systems (ICS), reference [32] develops a Convolutional Neural Network (CNN) based on Differential Evolution, eliminating the need for manual parameter tuning and achieving excellent performance. Reference [33] introduces a novel Graph Convolutional Neural Network (NE-GConv) that considers both node and edge features. Experimental results demonstrate that this method exhibits lower false positive rates and better computational efficiency compared to other Graph Neural Network (GNN) models. In reference [34], a new unsupervised approach is proposed, leveraging a Bidirectional Generative Adversarial Network (BiGAN) to detect anomalous behavior based on reconstruction errors in the feature space. Reference [35] combines the attention mechanism with a Convolutional Neural Network by incorporating attention mechanisms into the hierarchical layers of the network, effectively improving the detection rate for minority classes.

Random oversampling is a fundamental method for addressing imbalance problems. However, this approach has significant drawbacks as it achieves balance by randomly replicating minority class data, leading to overfitting issues during experimentation. The SMOTE algorithm [36,37] addresses the overfitting problem of random oversampling but suffers from the limitation of blindly selecting K-nearest neighbor data, making it challenging to determine the optimal sampling ratio, denoted as n. Borderline-SMOTE is an improved algorithm based on the SMOTE algorithm that focuses on sampling the boundaries of minority class data. It increases the number of minority class samples while minimizing the generation of noise and overlapping samples, thereby enhancing the performance of the classifier. Tomek Link and NCL [38,39,40] are two classical undersampling methods that partially address the class imbalance problem. However, it is difficult to determine the appropriate number of undersampled samples, and removing samples on the boundaries can alter the distribution of the original data, potentially lowering the performance of the classifier. Effectively addressing the class imbalance problem remains a major challenge in intrusion detection.

To effectively tackle the class imbalance problem, we propose a hybrid sampling method based on Borderline-SMOTE and GMM clustering. This method handles imbalanced data at the data level, avoiding changes to the distribution of the original data. Additionally, we utilize the QPSO algorithm to automatically obtain the parameters in the CNN, mitigating the limitations of manual parameter tuning. We have designed a data-driven intrusion detection model, BSGM-QOSO-1DCNN, and validated its effectiveness via experiments on the KDD99 dataset.

## 3. The Solution

This section describes the proposed BSGM-QPSO-1DCNN model, which combines the BSGM hybrid sampling technique and the QPSO algorithm. The model is illustrated in Figure 1. The model starts by preprocessing the original data, including numericalization, normalization, and one-hot encoding. Then, the original dataset is divided into a training set and a testing set. The training set undergoes BSGM hybrid sampling, while the testing set remains unchanged for model evaluation. Next, the QPSO algorithm is employed to automatically search for the optimal parameters within the CNN. Once the optimal parameters are obtained, the model is trained using the BSGM-processed training set. Finally, the model is tested using the unprocessed testing set.

### 3.1. Data Preprocessing

First of all, the original data have complex types, both character type data and numeric type data, and the CNN classifier can only handle numeric type data, so the original data need to be uniformly converted to numeric type data. For example, there are three types of Protocol_type, which represents the network protocol type, and the numerical results are shown in Table 1.

Secondly, the distribution of values of different categories of data is large in order to prevent the large difference in the data scale level from adversely affecting the deep learning model, the data need to be normalized to eliminate the data scale, and the normalized data range is between 0 and 1. The specific normalization method is shown in Equation (1).
(1)$$\chi_{i}^{,}=\frac{x_{i}-x_{min}}{x_{max}-x_{min}}\;$$

In the equation, $x_{i}$ and $x_{i}^{,}$ represent the original and normalized values, respectively, while $x_{min}$ and $x_{max}$ represent the minimum and maximum values of the data.

Finally, the attack types and normal types of the data are encoded using one-hot encoding.

### 3.2. The BSGM Hybrid Sampling Algorithm for Handling Imbalanced Problems

In practical applications, a common criterion often used is that a dataset is considered imbalanced when the minority class samples are less than 10% of the majority class samples. In this study, we used 10% of the entire KDD99 dataset as the original dataset, which consists of 494,016 data instances. In the KDD99 dataset, the quantities of Normal, DoS, Probe, R2L, and U2R data types are 97,278, 392,498, 4107, 86, and 52, respectively. The U2R data type accounts for only 0.01% of the entire dataset, significantly impacting the performance of the classifier. It is inappropriate to solely use oversampling or undersampling. Therefore, we propose a novel hybrid sampling algorithm called BSGM, which combines undersampling based on GMM clustering and oversampling using Borderline-SMOTE to balance the dataset.

First, calculate the mean value, denoted as $I_{mean}$, of the training set data *N* and data category *C*. The calculation formula is shown in Equation (2).
(2)$$I_{mean}=int\frac{N}{C}$$

For data types smaller than $I_{mean}$, Borderline-SMOTE is applied to perform oversampling, bringing the minority class data to a unified quantity. It is not a simple replication of data but rather identifies the K nearest neighbors for each minority class sample and categorizes them into pure minority class samples, borderline samples, and noise samples. Borderline samples refer to samples between the minority and majority classes, while noise samples are minority-class samples present within the majority-class data. Borderline-SMOTE only samples the borderline samples by calculating a synthetic factor based on the differences between the borderline samples and their K nearest neighbors. This synthetic factor is then multiplied by a weight coefficient, which is determined using the number of minority class samples among the K nearest neighbors of the borderline samples. Using the synthetic factor and weight coefficient, new samples are generated via linear interpolation.

For data types larger than $I_{mean}$, GMM is employed to cluster the data into C data clusters. A certain proportion of data is extracted from each data cluster, and a new dataset is synthesized. Finally, the data obtained from Borderline-SMOTE and GMM clustering undersampling are merged to form a balanced dataset. GMM is a probabilistic model that combines multiple single Gaussian distributions. It is widely used in tasks such as data clustering, anomaly detection, and density estimation. In GMM, each Gaussian distribution is called a component and is described with its mean, weight parameters, and covariance matrix. Data points can be assigned to different components, and the weight parameters of each component represent its importance in the entire model. Assuming there are K components, the probability density function of GMM can be written as follows:(3)$$p(x)=\sum_{k=1}^{K}\alpha_{k}N(x|\mu_{k},\Sigma_{k})\;$$
where $N(x|\mu_{k},\Sigma_{k})$ represents the value of Gaussian distribution, $\mu_{k}$ represents the average value, $\Sigma_{k}$ represents the covariance matrix, x represents the observed data, and $\alpha_{k}$ represents the weight parameter of the KTH component, satisfying the condition $\alpha_{k}\geq 0,\;\sum_{k=1}^{K}\alpha_{k}=1$.

The objective of the Gaussian Mixture Model (GMM) is to estimate the model parameters by maximizing the likelihood function of the observed data. This is typically achieved using the Expectation–Maximization (EM) algorithm. The EM algorithm is an iterative algorithm that alternates between two steps: the Expectation (E) step and the Maximization (M) step. In the E step, the posterior probabilities of each data point belonging to each component are computed. According to Bayes’ theorem, the posterior probability of data point xi belonging to the k-th component can be expressed as follows:(4)$${\;\gamma }_{ik}=\frac{\alpha_{k}N(\chi_{i}|\mu_{k},\Sigma_{k})}{\sum_{j=1}^{k}\alpha_{j}N(\chi_{i}|\mu_{k},\Sigma_{j})}\;$$

In the M step, these posterior probabilities are used to update the model parameters, particularly the weight parameters, means, and covariance matrices of each component. Finally, in the E step, the updated parameters are used, and the E and M steps are repeated until convergence. In GMM, the EM algorithm is often run multiple times with different random initial parameters, and the model with the highest likelihood value is selected as the final model. In summary, GMM is a flexible and powerful model that can be used for various data analysis and modeling tasks.

We only perform BSGM mixed sampling on the training set D = {Di, i = 1, 2, …, C}. If the data volume of Di is less than Imean, we use Borderline-SMOTE to oversample Di by a certain ratio, resulting in dataset Di′. If the data volume of Di is greater than Imean, we use GMM to cluster Di into C data clusters and then extract a portion of data from each cluster, merging them into dataset Di″. Finally, we combine dataset Di′ and dataset Di″ to obtain a balanced dataset D′. Algorithm 1 provides the pseudocode for the BSGM algorithm.
**Algorithm 1 BSGM Hybrid Sampling****Input:**Training set D = {Di, i = 1, 2, …, C};   C = the total number of classes;   N = the total number of training set;|Di|= the number of Di;**Output:**   a balanced training set D′
1: Imean=intNC
2:for i ← 1 to C do3:    if |Di|<Imean then4:      Di′=Borderline-SMOTE(Di,Di′)#Use Borderline-SMOTE to oversample Di
5:    end if6:    if |Di|>Imean then7:      Gk = GMM(Di,C)#Use GMM to cluster Di into C clusters8:      for k ← 1 to C do9:        Gk′ = Resample(Gk)10:      end for11:      Di″ = Concatenate(Gk′)12:    end if13:    D′ = Concatenate(Di′, Di′′)14:end for15:return D′


### 3.3. Classifier Based on CNN

A Convolutional Neural Network (CNN)-based classifier is designed in this paper to classify $D^{\prime }$. CNN is a feedforward neural network, which is mainly composed of a convolutional layer, pooling layer, fully connected layer, and output layer. The feature information of the input data is extracted via the convolution operation and pooling operation, and classification or regression is carried out via the fully connected layer. The convolutional layer uses convolution checks to carry out convolution operations on input data to obtain feature maps to extract feature information from input data. In the pooling layer, the number of parameters and calculation amount are reduced via the downsampling feature diagram, and the robustness of the model is enhanced. The fully connected layer is usually used at the last layer for the sorting or regression output.

The CNN structure designed in this paper is shown in Figure 2. The network structure consists of two convolution layers, two pooling layers, one Flatten layer, and two Dense layers. The activation function of the two convolutional layers is set to Relu, the activation function of the first Dense layer is set to Relu, and the activation function of the second Dense layer is set to Softmax. The convolution layer and the pooling layer are used to learn local features, and the Flatten layer is used to flatten the two-dimensional feature map into a one-dimensional feature vector. Finally, two Dense layers are used for feature recognition and classification.

### 3.4. Parameter Optimization Based on QPSO

CNN has a strong feature extraction ability, can handle massive high-dimension data, and has an end-to-end learning ability, which is very suitable for network intrusion detection. However, it is usually necessary to manually adjust the number of convolution kernels and other parameters when training the model, which is usually subjective and difficult to obtain the optimal parameters. QPSO is a particle swarm optimization algorithm based on quantum theory. It is different from traditional PSO in the aspects of using qubits as a state representation, particle updating formula containing random matrix, particle motion affected by system dynamics, dynamic reduced qubit weight, introduction of mutation operation, and realization of group cooperation. QPSO algorithm has a stronger global search ability and can eliminate the shortcomings caused by artificial parameter selection. Therefore, this paper uses the QPSO algorithm to search for parameters in CNN and uses mean square error (MSE) as the fitness function of the algorithm. The particle position update formula of the QPSO algorithm is as follows:(5)$$m_{best}=\frac{1}{m}\sum_{i=1}^{m}p_{best\_i}\;$$
(6)$$p_{i}=\varphi p_{best\_i}+(1-\varphi )p_{gest}\;$$
(7)$$X_{i+1}=p_{i}\pm \lambda |m_{best}-X_{i}|\ln\left(\frac{1}{u}\right)\;$$
where *m* represents the particle population size, $m_{best}$ represents the average value of the local optimal value of all particles, $p_{best_{-}i}$ represents the local optimal value of the ith particle in the current iteration, $p_{gest}$ represents the current global optimal solution of the population, φ represents the uniform distribution value on (0, 1), $X_{i}$ represents the position of the ith particle in the current iteration, $X_{i+1}$ represents the position of the ith particle in the next iteration, λ represents the innovation coefficient, which is the only control parameter in QPSO, and u represents the uniform distribution value on (0, 1).

We set the number of convolutional nuclei and batch size in CNN as the objects to be optimized, set the particle population as 30, the particle dimension equal to 3, set the search range of convolutional nuclei as [30, 120], and set the number of iterations as 10. The mean square error of the predicted results and the real results as the fitness function automatically outputs the optimal number of convolutional nuclei after 10 iterations. And batch size, the optimal parameters we obtain, are shown in Table 2.

## 4. Experimental Analysis

At the beginning of the experiment, we started by setting some hyperparameters. Firstly, using the Formulas (5)–(7) from Section 3.4, we calculated the optimal number of convolutional kernels in the CNN to be 55 and 99, as well as the optimal batch size of 110. We set these values, 55, 99, and 110, as the number of convolutional kernels and batch size for the BSGM-QPSO-1DCNN model, with an epoch set to 30. During the model training, we used a balanced dataset processed via BSGM. For the other models, the epoch was set to 30, and the batch size was set to 128. To validate the effectiveness of the proposed method, we conducted binary and five-class classification experiments for the BSGM-QPSO-1DCNN model and the comparative models on the KDD99 dataset.

We set the same parameters for ROS-CNN, SMOTE-CNN, RUS-SMOTE-CNN, and BCGM-QPSO-1DCNN models except for convolution kernel size and batch size, as shown in Figure 3. The convolution kernel size and step size of the two convolution layers are set to 3 and 1, and the activation function is set to Relu. kernel_number of the first convolution layer is set to 32, and kernel_number of the second convolution layer is set to 64. The Pool_size is set to 2, the step size to 2, and the activation function is set to Relu for both pooling layers. The activation function of the first Dense layer is set to Relu, and the activation function of the second Dense layer is set to Softmax.

### 4.1. Data Set Introduction

The KDD99 dataset is one of the most well-known datasets in the field of intrusion detection. It was collected by researchers at the MIT Lincoln Laboratory from July 1998 to August 1999. The dataset contains a large number of records from network traffic, including normal traffic and various types of attacks. NSL-KDD is an improved version of the KDD99 dataset that addresses some issues present in the original dataset, such as sample duplicates and redundancies. It also introduces new attack types, making the dataset more comprehensive and diverse. The UNSW-NB15 dataset is a network intrusion detection dataset released by the University of New South Wales. It contains a large amount of data from real-world network environments, including normal traffic and various types of attacks. The dataset aims to simulate intrusion activities in real networks and provide researchers with a dataset representing real-world scenarios. The CICIDS2017 dataset is an intrusion detection dataset released by a Canadian communication security company. It includes large-scale actual network traffic data, covering various attack types and normal traffic. It can be used to evaluate and improve the performance of intrusion detection systems in real network environments.

These datasets provide different types of network traffic data, including normal traffic and various attack types, and offer rich resources for researchers to develop and evaluate intrusion detection algorithms. The choice of an appropriate dataset depends on research requirements and specific areas of interest. The KDD99 dataset has issues such as class imbalance, duplicate samples, and redundant samples. NSL-KDD is an improved version of the KDD99 dataset that partially addresses the class imbalance issue. UNSW-NB15 and CICIDS2017 datasets are relatively newer and may lack relevant research and benchmark results compared to classic datasets like KDD99. Additionally, they may have higher demands for computing and storage resources. Therefore, we prioritize the use of the KDD99 dataset for our experiments. After achieving satisfactory results on this dataset, we will further validate our approach on other datasets.

In this study, we selected 494,016 data instances from the KDD99 dataset as the original dataset, with 412,728 instances used as the training set and 148,207 instances as the test set. The dataset is divided into five different categories: Normal, DoS, Probe, R2L, and U2R. Each connection is represented by 41 attributes, including source and destination IP addresses, port numbers, protocol types, etc. The original dataset is first preprocessed via numericalization, normalization, one-hot encoding, and other techniques. After data partitioning, BSGM mixed sampling is applied to the training set, while the test set remains unchanged. The sampled data are then fed into the model for training. The datasets used in this study are presented in Table 3.

### 4.2. Evaluation Metrics

When conducting network intrusion detection experiments, we often use four evaluation metrics to assess the overall performance of different models. These metrics include Recall, Precision, Accuracy, and F1-Measure, which are calculated based on the number of true positive, true negative, false positive, and false negative data. Each metric has its unique advantages and disadvantages, so we usually consider these metrics together in our experiments to assess the performance of the models.
(8)$$Recall=\frac{TP}{TP+FP}\;$$
(9)$$Precision=\frac{TP}{TP+TN}\;$$
(10)$$Accuracy=\frac{TP+TN}{TP+FP+TN+FN}$$
(11)$$F1\text{-}Measure=\frac{2*Precision*Reacll}{Precision+Reacll}\;$$

Among them, *TP* is true-positive data, predicted as attack data, and actual is also real attack data; *TN* is true-negative data, predicted as normal data, and actual is also real data, *FP* is false-positive data, predicted as attack data, and actual is normal data; and *FN* is false-negative data, predicted as normal data, and actual is attack data. Each indicator has its own characteristics, and the experimental results are generally evaluated by combining these indicators in the experiment.

### 4.3. Second Classification Experiment

In order to verify the effectiveness of the BSGM-QPSO-1DCNN model, we compared five models: ROS-CNN, SMOTE-CNN, RUS-SMOTE-CNN, RUS-SMOTE-RF, and RUS-SMOTE-MLP. The binary classification experiments were conducted on the KDD99 dataset, and the evaluation metrics were selected as Recall, Precision, Accuracy, and F1-Measure to evaluate the models. The number of convolutional kernels and batch size of our model are 55, 99, and 110, and the number of convolutional kernels and batch size of the comparison CNN model are 32, 64, 128, and other aspects we set the same parameters for ROS-CNN, SMOTE-CNN, RUS-SMOTE-CNN, and BSGM-QPSO-1DCNN models. The experimental results are shown in Table 4 and Table 5.

Table 4 shows the comparison experiments of the binary CNN models using different sampling methods on the KDD99 dataset, where the black bolded parts are the optimal values of a certain evaluation metric. We observe in Table 4 that the BSGM-QPSO-1DCNN model achieves 99.93%, 99.97%, 99.95, and 99.96% for Accuracy, Precision, Recall, and F1-Measure, respectively, with Recall and F1-Measure achieving the best classification results. Accuracy and Precision differed from the other models by 0.01%. Table 5 shows the comparison experiments of RF and MLP models with mixed sampling of RUS-SMOTE on the KDD99 dataset. We observe in Table 5 that the BSGM-QPSO-1DCNN model achieves the best classification results. Compared to the RUS-SMOTE RF model, our model Accuracy increased by 0.02%, Precision increased by 0.02%, Recall increased by 0.05%, and F1-Measure increased by 0.02%. Compared with the RUS-SMOTE MLP model, the Accuracy, Precision, Recall and F1-Measure of our model were increased by 0.02%, 0.02%, 0.03% and 0.02% respectively.

Our model is inferior to ROS-CNN in Accuracy and Precision, which we believe is due to the overfitting of the duplicate data after the ROS sampling process. The model achieves optimal results in Recall and F1-Measure. Comparing the RF and MLP models, the overall performance of our model is optimal.

### 4.4. Multi-Category Experiments

To further validate the effectiveness of the BSGM-QPSO-1DCNN model, we compared five models, namely, the ROS-CNN model, SMOTE-CNN, RUS-SMOTE-CNN, RUS-SMOTE-RF, and RUS-SMOTE-MLP. Multi-classification experiments were conducted on the KDD99 dataset, and the evaluation metrics were selected as Recall, Precision, Accuracy, and F1-Measure to evaluate the models. The parameters of the CNN comparison model, RF, and MLP comparison models were set unchanged. The experimental results are shown in Table 6, Table 7, and Table 8.

Table 6 shows the comparative experiments of multiclassification CNN models using different sampling methods on the KDD99 dataset, where the black bolded parts are the optimal values of a certain evaluation metric for a certain class. We observe in Table 6 that the BSGM-QPSO-1DCNN model achieves 99.94%, 99.94%, 99.94, 99.94, and 99.94% for Accuracy, Precision, Recall, and F1-Measure, respectively, which are the best classification results compared with the comparison model. Compared with RUS-SMOTE-CNN, Accuracy improved by 0.13%, Precision improved by 0.04%, Recall improved by 0.13%, and F1-Measure improved by 0.09%. Table 7 shows the multi-classification comparison experiments of RF and MLP models using RUS-SMOTE hybrid sampling on the KDD99 dataset, where we observe that the BSGM-QPSO-1DCNN model achieves the best classification results. Our model Accuracy improved by up to 0.36% compared to the RUS-SMOTE RF and RUS-SMOTE MLP models. Compared to the RUS-SMOTE RF model, our model Precision increased by 0.20%, Recall increased by 0.36%, and F1-Measure increased by 0.29%. The experimental results in Table 6 and Table 7 validate the effectiveness of our model for multi-classification experiments on the KDD99 dataset.

Table 8 shows the performance evaluation of each class for the multiclassification experiments on the KDD99 dataset, where the black bolded parts are the optimal values of a certain evaluation metric for a particular class. The results in the table clearly show the significant improvement in Precision and F1-Measure of U2R and R2L after the class imbalance treatment. Compared with the RF model, our model improves the Precision of U2R and R2L from 10% and 10% to 78% and 76%, and the F1-Measure from 18% and 19% to 78% and 76%. The best results were obtained for the RF model in Accuracy, the best results for Precision and F1-Measure, and overall, the best results were obtained for the performance evaluation of each class by our model.

Finally, we also plot the confusion matrices of six models in the multi-classification experiments, namely ROS-CNN, SMOTE-CNN, RUS-SMOTE-CNN, RUS-SMOTE-RF, RUS-SMOTE-MLP, and BSGM-QPSO-1DCNN. The plots of these confusion matrices further validate the effectiveness of our models, and their confusion matrices are shown in Figure 4.

From Figure 4f, it can be clearly seen that our model has only 26 false positives in the normal class with a false positive rate of 0.08%, while the false positive rates of other models are 0.14% (Figure 4a), 0.15% (Figure 4b), 0.22% (Figure 4c), 0.81% (Figure 4d), and 0.34% (Figure 4e). Figure 4b,f shows that the SMOTE-CNN model and our model have similar false positives on R2L, which is the best result among all models. It is obvious from Figure 4f that our model has only four false positives on U2R with 22% false positives, and the false positives of other models are 87% (Figure 4a), 84% (Figure 4b), 90% (Figure 4c), 89% (Figure 4d), and 87% (Figure 4e). Our model significantly reduces the false alarm rate of U2R. On Dos, Probe, and R2L classes, our model also achieved lower false alarm rates. In addition, from Figure 4f, we can see that our model has the lowest false alarm rate of 0.17% in the normal class, while the false alarm rates of other models are 0.22% (Figure 4a), 0.23% (Figure 4b), 0.32% (Figure 4c),0.39% (Figure 4d), and 0.23% (Figure 4e). Figure 4f shows that our model has the lowest false alarm rate on the Dos class has the lowest miss rate of 0.01%; our model also achieves lower miss rates on Dos, Probe, and R2L classes. Combining the leakage and false alarm rates, it can be seen from Figure 4 that the overall performance of our model is optimal and effectively reduces the false alarm and leakage rates for a few classes, further validating the effectiveness of our proposed model.

## 5. Conclusions

We propose a new network intrusion detection model, BSGM-QPSO-1DCNN, which combines class imbalance processing techniques and automatic parameter optimization. To deal with the class imbalance problem, we design a hybrid sampling technique BSGM that combines Borderline-SMOTE and GMM clustering undersampling. To eliminate the deficiencies caused by human tuning, we use the QPSO algorithm to automatically find the parameters in the CNN.

Our model achieves the best overall performance in both binary and multiclassification experiments on the KDD99 dataset with ROS-CNN, SMOTE-CNN, RUS-SMOTE-CN, RUS-SMOTE-RF, and RUS-SMOTE-MLP. In particular, in the multi-classification experiments, our model improves the Precision of U2R and R2L from 10% and 10% to 78% and 76%, and F1-Measure from 18% and 19% to 78% and 76%, which fully verifies the effectiveness of our model. The class imbalance problem does not only exist in the field of intrusion detection but also in many academic studies. For example, cancer genetic testing data, and telecommunication fraud detection. In the future, we will validate the model on other datasets and apply the model to other fields.

To further validate our model, we will conduct experiments on the UNSW-NB15 and CICIDS2017 datasets. Different datasets have different characteristics and distributions, and testing on multiple datasets can provide more comprehensive and accurate evaluation results, helping to determine the model’s performance and practicality, thereby improving the model and enhancing its applicability.

Furthermore, we also consider real-time performance in our future work. Real-time capability is an important attribute of intrusion detection, emphasizing the system’s ability to promptly detect and respond to intrusion intruders. By using real-time intrusion detection, security teams can quickly identify the activities of intruders, take appropriate countermeasures, prevent further attacks, and mitigate potential damages and risks. We will create a simulated real-time environment to evaluate the model’s real-time performance. Based on the model’s performance in the simulated environment, we will propose improvements and optimizations to enhance the model’s real-time performance.

The issue of class imbalance is not only present in the field of intrusion detection but also widely encountered in various academic research domains, such as cancer gene detection data and telecom fraud detection. Different domains have different characteristics and requirements. In the future, we will perform corresponding data preparation to adapt the convolutional neural network as a classifier to the specific needs of each domain and apply the model to other fields.

## Figures and Tables

**Figure 1 sensors-23-06206-f001:**
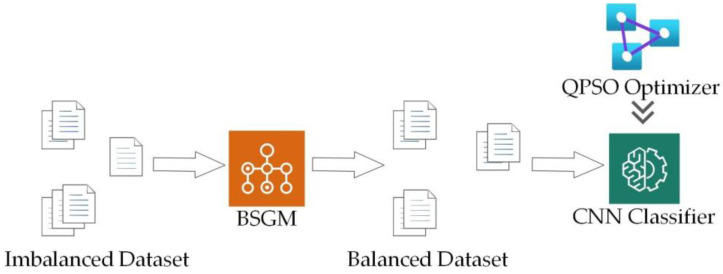
Structure diagram of BSGM-QPSO-1DCNN model.

**Figure 2 sensors-23-06206-f002:**
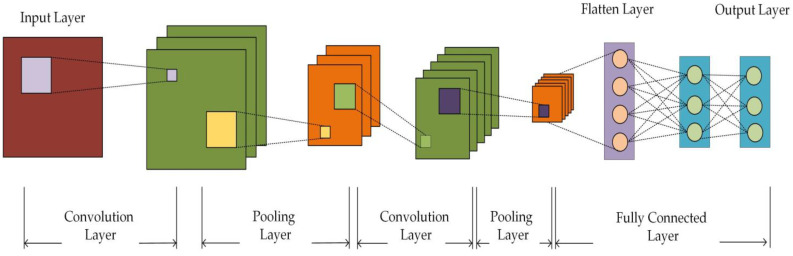
CNN model structure diagram.

**Figure 3 sensors-23-06206-f003:**
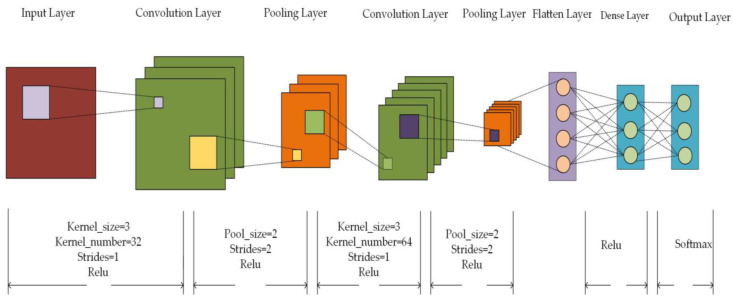
Parameter diagram of CNN model.

**Figure 4 sensors-23-06206-f004:**
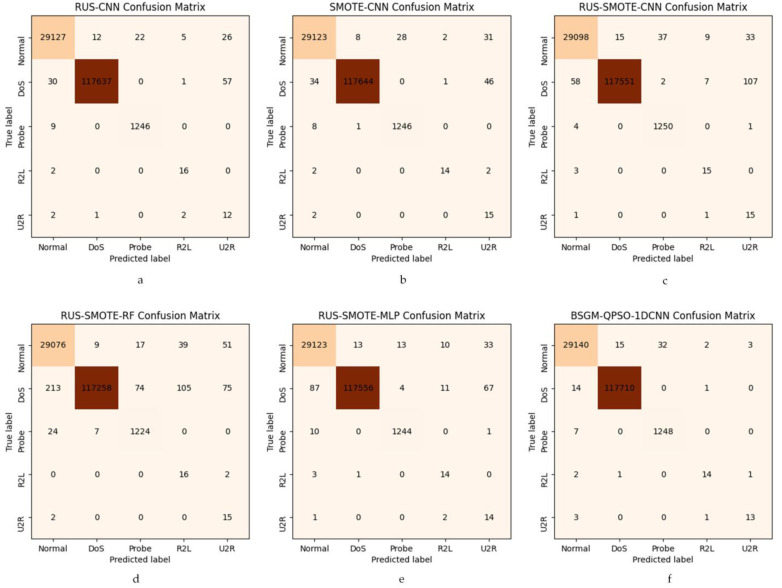
Confusion matrix of each model.

**Table 1 sensors-23-06206-t001:** Numerical Protocol_type data.

Protocol_type	icmp	tcp	udp
coding	1	2	3

**Table 2 sensors-23-06206-t002:** Optimal parameters of the BSGM-QPSO-CNN model.

Model	First_kernel_num	Second_kernel_num	Batch_size
BSGM-QPSO-1DCNN	55	99	110

**Table 3 sensors-23-06206-t003:** Sample number of each class in KDD99 data set.

Class	Trainset-size	Testset-size	Total
Normal	68,086	29,192	97,278
Dos	274,773	117,725	392,498
Probe	2852	1255	4107
R2L	68	18	86
U2R	35	17	52
Total	412,728	148,207	494,016

**Table 4 sensors-23-06206-t004:** Performance evaluation of CNN model for binary classification on KDD99 dataset (%).

Model	Accuracy	Precision	Recall	F1-Measure
ROS-CNN	**99.94**	**99.98**	99.94	99.96
SMOTE-CNN	99.94	99.98	99.94	99.96
RUS-SMOTE-CNN	99.93	99.98	99.93	99.96
BSGM-QPSO-1DCNN	99.93	99.97	**99.95**	**99.96**

**Table 5 sensors-23-06206-t005:** Performance evaluation of RF and MLP models for binary classification on the KDD99 dataset (%).

Model	Accuracy	Precision	Recall	F1-Measure
RUS-SMOTE-RF	99.91	99.95	99.90	99.94
RUS-SMOTE-MLP	99.91	99.97	99.92	99.94
BSGM-QPSO-1DCNN	**99.93**	**99.97**	**99.95**	**99.96**

**Table 6 sensors-23-06206-t006:** Performance evaluation of CNN model for multiple classifications on KDD99 dataset (%).

Model	Accuracy	Precision	Recall	F1-Measure
ROS-CNN	99.88	99.93	99.88	99.90
SMOTE-CNN	99.89	99.93	99.88	99.91
RUS-SMOTE-CNN	99.81	99.90	99.81	99.85
BSGM-QPSO-1DCNN	**99.94**	**99.94**	**99.94**	**99.94**

**Table 7 sensors-23-06206-t007:** Performance evaluation of RF and MLP models for multiple classification on KDD99 dataset (%).

Model	Accuracy	Precision	Recall	F1-Measure
RUS-SMOTE-RF	99.58	99.74	99.58	99.65
RUS-SMOTE-MLP	99.93	99.93	99.93	99.93
BSGM-QPSO-1DCNN	**99.94**	**99.94**	**99.94**	**99.94**

**Table 8 sensors-23-06206-t008:** Performance evaluation of RF and MLP models on KDD99 dataset for each class in multiclassification (%).

Model	Class	Accuracy	Precision	Recall	F1-Measure
RUS-SMOTE-RF	Normal	100	99	100	99
	DoS	100	100	100	100
	Probe	98	93	98	95
	R2L	**89**	10	**89**	18
	U2R	**88**	10	**88**	19
RUS-SMOTE-MLP	Normal	100	100	100	100
	DoS	100	100	100	100
	Probe	99	**99**	99	**99**
	R2L	78	38	78	51
	U2R	82	12	82	21
BSGM-QPSO-1DCNN	Normal	**100**	**100**	**100**	**100**
	DoS	**100**	**100**	**100**	**100**
	Probe	**99**	97	**99**	98
	R2L	78	**78**	78	**78**
	U2R	76	**76**	76	**76**

## Data Availability

The data used to support the findings of this study can be downloaded from https://archive.ics.uci.edu/ml/datasets/kdd+cup+1999+data.

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
