# Peer review of "Research on Adaptive 1DCNN Network Intrusion Detection Technology Based on BSGM Mixed Sampling"

_sensors, 2023, doi:10.3390/s23136206_

Round 1

Reviewer 1 Report

The idea of combining several aproaches (over and under sampling) seems rather interesting. In general article is of good quality, follows the scientific style. 

There are just several remarks, that should be addressed: 

1. In Into/Related work section, only CNN application is mentioned, while other ML/DL methods are not analyzed. This should be either corrected or concentration on just CNN motivated.

2. Rather similar comment related to the use of KDD99 dataset. It is really old dataset and there lot of new ones. Understandable, that for the task being solved it is not critical, since here a general approach, rather than metrics achieved are important, but motivation for choosing KDD99 should be given, as well as short review of existing dataset (like of CIC) should be given.

English language is fine. 

Reviewer 2 Report

The goal of this paper, as exposed by the authors, is to present a hybrid sampling tech-nique called Borderline-SMOTE and GMM, referred to as BSGM, which combines the two approaches.

Sections 1 and 2 contain a well-structured introduction and an in-depth review of state of the art. The authors successfully describe problems related to network intrusion detection, ML, mixed sampling, etc.

Figure 1 is difficult to follow with the text oriented both horizontally and vertically, although the authors had enough space to make it more reader-friendly and clearer from a technical point of view. This figure (or the next one) must be representative for an article published in a prestigious journal such as Sensors.

Additional, the authors should summarize their main contributions in this study in bullets in the end of the Introduction section. For each point mentioned in the contribution paragraph, identify which part in the resubmitted manuscript considers that point.

The authors should discuss the research gap and existing problems in this step as the research motivation.

INDEX TERMS need to be revised because for example "Mixed sampling" appear only in the Keywords list and sub-section 4.3. Demonstrate and explain the real-time mixed sampling capabilities of the proposed system in the intrusion detection field.

The experimental data are presented and analyzed in detail. Where is equations 5-7 applied to obtain the experimental results from chapter 4?

Section V is too short and concise. Perhaps a discussion section would be welcome and technically useful to complete the article.

The reference section is weak, citing few new and relevant articles in the research area.

Round 2

Reviewer 2 Report

The paper was improved by the revision process.

The authors have addressed most of my concerns satisfactorily.